# CT-Detected MTA Score Related to Disability and Behavior in Older People with Cognitive Impairment

**DOI:** 10.3390/biomedicines10061381

**Published:** 2022-06-10

**Authors:** Michele Lauriola, Grazia D’Onofrio, Annamaria la Torre, Filomena Ciccone, Carmela Germano, Leandro Cascavilla, Antonio Greco

**Affiliations:** 1Complex Unit of Geriatrics, Department of Medical Sciences, Fondazione IRCCS Casa Sollievo della Sofferenza, 71013 San Giovanni Rotondo, Italy; m.lauriola@operapadrepio.it (M.L.); c.germano@operapadrepio.it (C.G.); l.cascavilla@operapadrepio.it (L.C.); a.greco@operapadrepio.it (A.G.); 2Clinical Psychology Service, Health Department, Fondazione IRCCS Casa Sollievo della Sofferenza, 71013 San Giovanni Rotondo, Italy; f.ciccone@operapadrepio.it; 3Laboratory of Gerontology and Geriatrics, Fondazione IRCCS Casa Sollievo della Sofferenza, 71013 San Giovanni Rotondo, Italy

**Keywords:** atrophy, cognitive impairment, dementia, functional status, neuropsychiatric symptoms, homocysteine level

## Abstract

Our study aims to investigate the relationship between medial temporal lobe atrophy (MTA) score, assessed by computed tomography (CT) scans, and functional impairment, cognitive deficit, and psycho-behavioral disorder severity. Overall, 239 (M = 92, F = 147; mean age of 79.3 ± 6.8 years) patients were evaluated with cognitive, neuropsychiatric, affective, and functional assessment scales. MTA was evaluated from 0 (no atrophy) to 4 (severe atrophy). The homocysteine serum was set to two levels: between 0 and 10 µmol/L, and >10 µmol/L. The cholesterol and glycemia blood concentrations were measured. Hypertension and atrial fibrillation presence/absence were collected. A total of 14 patients were MTA 0, 44 patients were MTA 1, 63 patients were MTA 2, 79 patients were MTA 3, and 39 patients were MTA 4. Cognitive (*p* < 0.0001) and functional (*p* < 0.0001) parameters decreased according to the MTA severity. According to the diagnosis distribution, AD patient percentages increased by MTA severity (*p* < 0.0001). In addition, the homocysteine levels increased according to MTA severity (*p* < 0.0001). Depression (*p* < 0.0001) and anxiety (*p* = 0.001) increased according to MTA severity. This study encourages and supports the potential role of MTA score and CT scan in the field of neurodegenerative disorder research and diagnosis.

## 1. Introduction

Dementia is now one of the main causes of functional deficit in the elderly, with a progressive increase in the need for assistance and care. Besides, the number of people affected by dementia is likely to reach 82 million in 2030 and 152 million in 2050 [1]. Alzheimer’s disease (AD) represents about 60–70% of dementia cases [1] among western countries. Vascular dementia (VaD) represents about 5% to 10% of all dementia cases [2]. In recent years, diagnostic protocols have undergone an enormous evolution because of the development of new biomolecular diagnostic and imaging technologies. In addition to cognitive, behavioral, and functional assessments, we have very sensitive and specific diagnostic tools available that enable the early diagnosis and treatment of AD and other causes of dementia. AD is a neurodegenerative disease related to progressive cognitive, behavioral, and functional impairments that features the following underlying pathological signs: extracellular amyloid beta (A*β*) plaques and the accumulation of intracellular neurofibrillary tangles (NFTs) [3]. AD follows a progressive disease continuum, extending from an asymptomatic phase with evidence of AD biomarkers (preclinical AD), through minor (mild cognitive impairment (MCI)) and/or neurobehavioral (mild behavioral impairment (MBI) changes) to dementia AD [4,5,6]. Important biomarker information can be gleaned from imaging modalities such as magnetic resonance imaging (MRI), computed tomography (CT), and positive emission tomography (PET) that visualize early structural and molecular changes in the brain [7,8]. Fluid biomarker tests, such as cerebrospinal fluid (CSF), may also be used; CSF biomarkers can directly reflect the presence of *Aβ* and aggregated tau within the brain [4,9]. Given the growing number of cases and the imminent need to assist numerous patients, it is equally necessary that diagnostic and screening techniques are widely present in all territories and that they are economically sustainable.

MRI and CT scans have specific value as widely available and non-invasive examination techniques. They allow us to study the global cortical atrophy, medial temporal lobe atrophy, and white matter change evaluation. Moreover, there is sufficient evidence to suggest that MRI and CT have the same diagnostic accuracy with respect to identifying cerebral atrophy and cerebrovascular changes in autopsy-confirmed and clinical cohorts of VaD, AD, and mixed dementia [10,11]. In our clinic, all patients first undergo brain CT to study the morphology and degree of brain atrophy, with great attention to the extent of vascular damage, and we calculate the medial temporal lobe atrophy (MTA) score. Next, we decide if brain MRI and PET scan needed. The MTA score, published by Scheltens and colleagues in 1992 [12], is a simple measure by which mesio-temporal atrophy can be quantified. MTA is assumed to reflect disease severity because it is related to the burden of AD [13,14]. Volumetric studies of these areas show a 10% decrease in volume in patients with MCI due to AD [15,16] and a correlation of these volumes with the severity of cognitive impairment [17]. Consistently, hippocampal atrophy and parietal atrophy are significant predictors of progression from MCI to dementia, most often due to AD [18]. Recent studies indicate that medial temporal lobe atrophy potentially contributes to the etiology of behavioral and psychological symptoms and provide evidence to support the hypothesis that Alzheimer’s disease itself may contribute to the onset of these disorders [19].

Our study aims to confirm the association between the MTA score from CT scans and the severity of cognitive deficits and to deepen the relationship with the presence of psycho-behavioral disorders, which are known to be a major cause of disability. We analyzed in this population the levels of homocysteine, which is known as a vascular risk factor for cognitive impairment and disability [20]. The main purpose, therefore, is to investigate the possibility of obtaining a relationship between the MTA score and the severity of the deficit in the basal and instrumental activities of daily living.

## 2. Materials and Methods

This study was conducted according to the Declaration of Helsinki, the Guidelines for Good Clinical Practice, and the Guidelines for Strengthening the Reporting of Observational Studies in Epidemiology and was approved by the local ethics committee for human trials (Prot. No. 3877/DS). It was an observational study in which the assignment of an intervention to participants, its assessment of effects, and biomedical or behavioral health-related outcomes were not considered. Healthy participants were recruited as control subjects.

### 2.1. Study Sample

From November 2019 to October 2021, older subjects with cognitive impairment (CI) who had consecutively attended the Assessment Unit of Cognitive Impairment of the Complex Unit of Geriatrics of the Institute of Hospitalization and Scientific Care (IRCCS) “Casa Sollievo della Soflievo” were selected for the possible enrollment in the study. Control group patients were recruited into the geriatric unit and were classified as patients without cognitive impairment (NoCI) by cognitive, neuropsychiatric, and functional assessments. We obtained written informed consent for the research from each patient or from relatives or legal guardians. All subjects were Caucasians, excluding people of Jewish, eastern European, or north African descent, with most of the people having southern Italian origins, having lived in southern Italy for at least three generations. The inclusion criteria were: (1) age ≥ 50 years; (2) diagnosis of MCI according to the National Institute on Aging-Alzheimer’s Association (NIAAA) [21] criteria; (3) diagnosis of AD according to the NIAAA [22] criteria; (4) VaD diagnosis according to the criteria of the Working Group of the National Institute of Neurological Disorders and Stroke-Association Internationale pour la Recherche et l’Enseignement en Neurosciences (NINDS-AIREN) [23]; (5) the ability to provide informed consent or the availability of a relative or legal guardian in the case of patients with severe dementia. The exclusion criteria were: (1) the presence of serious comorbidities, tumors, other diseases, or physiological status (ascertained blood infections and thyroid, kidney, or liver disorders), that could be causally related to cognitive impairment and (2) a history of alcohol or drug abuse, head trauma, and other causes that can cause memory damage.

As a control group, we included older patients that were evaluated consecutively in the same center who did not have cognitive impairment or neuropsychiatric symptoms.

### 2.2. Clinical, Cognitive, Neuropsychiatric, and Functional Assessment

The patient medical status was gathered from a structured interview, clinical evaluation, and a review of patient primary care records.

In all patients, cognitive status defined with the Mini-Mental State Examination (MMSE) [24] and the Frontal Assessment Battery (FAB) [25] after a brief interview with the caregiver. The differential diagnosis between AD and VaD also relied on the Hachinski Ischemic Score (HIS) [26] to address unclear diagnoses of AD/VaD.

Neuropsychiatric symptoms were assessed by the Neuropsychiatric Inventory (NPI) [27]. It consists of 12 neuropsychiatric domains (delusions, hallucinations, depression, anxiety, agitation/aggression, euphoria, disinhibition, irritability/lability, apathy, motor aberration, physical activity, sleep disorders, and eating disorders).

In all patients, the functional status was assessed by the Activity of Daily Living (ADL) [28] and Instrumental Activities of Daily Living (IADL) [29] scales.

### 2.3. MTA Score Detection

A diagnosis of dementia was always supported by neuroimaging evidence. All patients underwent a brain non-contrast CT. The instrument used was a Toshiba Aquilion 64-slice CT (Manufacture product code: CTSCAN74442-3493, Japan). The slice thickness was 10 mm. Radiograms were evaluated by trained radiologists who were not informed about the clinical characteristics of the patients. In particular, by CT scans, the presence of multiple cortical/subcortical infarcts or an infarct in a strategic area such as the thalamus or temporal lobe and/or lesions of the white matter indicated probable VaD; the absence of the above-mentioned cerebrovascular lesions indicated AD. Using the choroidal fissure width, temporal horn, and hippocampal formation height, atrophy was assessed in five MTA degrees (0 to 4) [12]. A score of 0 indicated no atrophy, a score of 1 indicated an enlargement of the choroidal fissure, a score of 2 included a further enlargement of the temporal horn of the lateral ventricle, and the height of hippocampal formation was slightly reduced, a score of 3 included a moderate volume loss of the hippocampal formation, and a score of 4 indicated an increase in all of these outcomes in the final phase. 

### 2.4. Quantification of Homocysteine and Other Biochemical Concentrations

Blood samples (3–5 mL of blood) were collected intravenously from all upper limbs of the patients in the morning. Then, the blood samples were stored in Vacutainer tubes containing citrate; within no more than 30 min, the samples were transferred to the biochemistry department and analyzed in a full-auto analyzer machine. 

The serum homocysteine was set to two levels: a homocysteine level between 0 and 10 µmol/L and a homocysteine level > 10 µmol/L.

The blood concentrations of cholesterol and glycemia were measured in all patients.

Moreover, the presence/absence of hypertension and atrial fibrillation was collected.

### 2.5. Statistical Analyses

For dichotomous variables, hypotheses regarding differences between groups were tested using the chi-squared test. This analysis was performed using a 2-way contingency table analysis. For continuous variables, the normal distribution was verified by the Shapiro–Wilk test of normality and the one-sample Kolgomorov–Smirnov test. For normally distributed variables, the assumptions regarding the differences between the groups were compared by the two-sample Welch *t*-test or by the analysis of variance (ANOVA) in the general linear model. For variables that were not normally distributed, the assumptions for differences between the groups were compared by the Wilcoxon rank sum test with a continuity correction or by the Kruskal–Wallis rank sum test. Risks (adjusted for the presence/absence and severity of cognitive impairment) were reported as odds ratios (OR) together with their 95% confidence intervals (CI). All statistical analyses were performed with the R Ver. 2.8.1 statistical software package (The R Project for Statistical Computing; available at URL http://www.r-project.org/ (accessed on 3 July 2021). Tests in which the *p*-value was less than the type I error rate of α = 0.05 were declared significant.

## 3. Results

In the course of the enrolment period, 250 older patients were screened for inclusion in the study. Of these, seven patients were excluded because they were younger than 50 years, and four patients had incomplete examinations. Therefore, the final population included 239 patients, 92 men (38.5%) and 147 women (61.5%) with a mean age of 79.29 ± 6.84 years and a range from 53 to 97 years. As explained in Table 1, according to the MTA score, 14 patients were MTA 0, 44 patients were MTA 1, 63 patients were MTA 2, 79 patients were MTA 3, and 39 patients were MTA 4.

The groups differed in sex distribution: females were present in MTA 2–4 (*p* = 0.025). The mean age increased according to the MTA severity (*p* < 0.0001).

The cognitive and functional parameters decreased according to the MTA severity: MMSE (*p* < 0.0001), FAB (*p* = 0.017), ADL (*p* < 0.0001), and IADL (*p* < 0.0001). Moreover, according to diagnosis distribution, AD patient percentages increased by MTA severity (*p* < 0.0001), while no/mild cognitive impairment patient percentages tended to decrease.

In addition, homocysteine scores increased according to the MTA severity (*p* = 0.032) as well as the homocysteine level (homocysteine level between 0 and 10 µmol/L and homocysteine level > 10 µmol/L) distribution, as shown in Figure 1 (OR = 1.963; 95% CI = 1.705–2.219; *p* < 0.0001).

The patient groups did not differ in hypertension (*p* = 0.996), diabetes (*p* = 0.450), atrial fibrillation (*p* = 0.735), or dyslipidemia (*p* = 0.404).

In Figure 2, the distributions of the MTA score according to the cognitive impairment presence/absence and severity are shown. CI patients showed a higher frequency of MTA 4 (OR = 21.696; 95% CI = 19.613–23.779; *p* < 0.0001), than the NoCI group. According to the cognitive impairment severity, patients with moderate and severe dementia had a higher frequency of MTA 2–4 (OR = 19.899; 95% CI = 17.715–22.083; *p* < 0.0001) than MCI and mild dementia. 

In Figure 3, the MTA severity was directly proportional to increased functional impairment severity for ADL (OR = 4.996; 95% CI = 4.537–5.456; *p* < 0.0001) and IADL (OR = 5.553; 95% CI = 4.668–6.437; *p* < 0.0001), respectively.

In Table 2, the association of the MTA score and NPI domains, adjusted by cognitive impairment presence/absence and severity, is shown. Depression and anxiety increased according to the MTA severity (NPI-Depression, OR = 4.869; 95% CI = 3.053-6.686; *p* < 0.0001 and NPI-Anxiety, OR = 2.792; 95% CI = 1.120–4.464; *p* = 0.001), while the other NPI domains did not show significant differences.

## 4. Discussion

Nowadays, cognitive and functional impairment represents a major concern for elderly people, and it is expected to severely impact the public health and medical care system in the coming decades, mainly due to the increase in life expectancy in both developed and developing countries [30,31,32]. The ultimate goal of basic and clinical research is to reduce the burden of cognitive and functional impairment as we age. Intense work is underway to improve the clinical management of affected people and reduce the incidence of the diseases that cause cognitive and functional impairment [33].

Cognitive and functional impairment is a typical feature of neurodegenerative disorders, including AD [34]. In the diagnostic setting of AD, brain imaging moved from a minor to central role [35,36].

In fact, given the inaccessibility of the brain, imaging has a key role as a “window on the brain” being able to detect the signs of structural and functional cerebral alteration in symptomatic individuals with high specificity and allowing prognostication. Moreover, imaging may support differential diagnosis by identifying alternative and/or contributory pathologies [35].

CT before and [37] after, structural and functional MRI, PET, studies of cerebral metabolism with fluoro-deoxy-D-glucose (FDG), and amyloid tracers such as Pittsburgh Compound-B (PiB) [22] have been used in the diagnosis of AD [35]. 

The MTA visual rating, proposed by Scheltens et al. in 1992 [12], was validated for AD patients and then was incorporated in clinical criteria [38]. 

The MTA visual rating could be used on CT brain scans [39] with an accuracy similar to MRI [12,40]. 

Studies have shown that in AD and MCI an atrophy of the entorhinal cortex and hippocampus is observed on MRI and CT. A recent study has shown that hippocampal atrophy, measured on MRI, would have an additive value in the diagnostic accuracy of AD [41]. The researchers concluded that measuring the hippocampus, particularly the hippocampal atrophy index, best discriminates those with MCI from cognitively healthy people, suggesting that the hippocampal atrophy value is the most sensitive index in the early stages of the disease and that the combination of the baseline hippocampal volume and the atrophy index best predicts the conversion from MCI to AD. On the contrary, the total cortical atrophy index seems to better distinguish between AD and MCI, making their use more advantageous in the advanced stages of the disease. Another recent study reported that MTA and parietal atrophy are specific for AD, while asymmetric frontal lobe atrophy, temporal pole atrophy, and an anterior-to-posterior gradient of atrophy are specific for FTD [42]. In the same study, when compared to AD and VaD, MTA values were strongly associated with cognitive function in both groups [42]. Currently, a difference in MTA score performance in CT compared to MRI has not been demonstrated [43].

In this framework, we assessed the MTA score using CT scan in a cohort of 239 elderly outpatients that attended to the Cognitive Impairment Evaluation Unit of the Complex Unit of Geriatrics, Fondazione Casa Sollievo della Sofferenza-IRCCS, San Giovanni Rotondo, Italy.

The results showed a decrease in the cognitive and functional parameters according to the MTA severity: MMSE (*p* < 0.0001), FAB (*p* = 0.017), ADL (*p* < 0.0001), and IADL (*p* < 0.0001). For diagnosis distribution, AD patient percentages increased according to MTA severity (*p* < 0.0001), unlike no/mild cognitive impairment patient percentages. These results are consistent with other studies that proved the role of MTA evaluation as a biomarker for Alzheimer’s disease (AD) and as a useful parameter associated with memory deficits [39,44]. 

As a matter of fact, only a few studies showed that MTA scales are applicable to CT in clinical practice [40,41,42,43,44,45,46,47], and even fewer demonstrated the use of MTA score by CT scan in diagnosing AD [36,45,47,48].

Rather, several studies demonstrated that, by using MRI, the MTA was present in patients with mild cognitive impairment [49,50,51] and was able to differentiate between AD patients and controls [52,53,54].

Impairment in ADL is a major problem in AD and it is related to increased caregiver burden [55]. By using MRI, some studies revealed that brain atrophy influences functional ability [55,56,57]. Based on our literature review, our study is the first to correlate functional (ADL and IADL) scales and MTA score using CT imaging.

Depression is a common comorbidity that is seen in AD and in mild cognitive impairment [58]. Depression and AD may have common etiological mechanisms. Depression may cause increased circulation of glucocorticoids, which, in turn, could lead to hippocampal atrophy [59]. By using MRI, one study has reported that patients with depression in AD had more MTA than patients with AD without depression [60]. Our results, by using CT scan, ascertained alterations in psycho-behavioral profiles according to MTA severity, with the tendency to manifest depression (*p* < 0.0001) and anxiety (*p* = 0.001). These findings provide support for the idea that regional atrophy in different brain structure may be intimately related to cognitive and non-cognitive symptoms. As reported by García-Alberca et al., MTA is potentially associated to the etiology of psycho-behavioral symptoms in AD [19].

The brain undergoes cerebral atrophy with aging. This occurs in both healthy and pathological aging. In this study, we also evaluated the relationship between mean age and MTA scores. We observed that the mean age positively correlated with MTA severity (*p* < 0.0001). These data are in line with the assumption that age-related brain atrophy is strongly associated to neurodegenerative diseases such as AD [61,62,63], and retracing the visual MTA assessment using CT scans is a feasible and decade-specific tool in the management of elderly patients with memory deficits [45]. 

Raised plasma homocysteine is an established risk factor for cardiovascular disease, dementia, cognitive impairment, and mortality [64]. In our previous work, we identified a link between cognitive impairment, functional complications, psycho-behavioral alterations, and homocysteine plasma levels [65]. In this work, we found an increase in homocysteine scores (*p* = 0.032) and its level distribution (*p* < 0.0001) according to MTA severity.

However, our observational study has some limitations. First, it is limited by a small sample size, as the study population was recruited by a single institution. Second, it included only Caucasian patients. Thus it is possible that our results may not be confirmed in other ethnicities. Third, CSF biomarkers were lacking. These observations lay the groundwork for a multicentric evaluation for standardizing the method.

## 5. Conclusions

In conclusion, suggesting a combination of a method of simple visual rating (MTA score) and a faster imaging technique by CT scan, the findings of this observational study encourage and support the potential role of CT in the field of neurodegenerative disorder research and diagnosis. In fact, although CT is far less sensitive than the other imaging techniques, it may still have a role in detecting changes associated with cognitive impairment, in particular considering the possible advantages attributable to the lower cost, the shorter acquisition time, being more adaptable to patients with poor compliance, and the possibility that it can be performed in patients with ferromagnetic implants or cardiostimulators. The MTA score could be considered a radiological biomarker into a multidimensional evaluation of the patients with cognitive impairment.

## Figures and Tables

**Figure 1 biomedicines-10-01381-f001:**
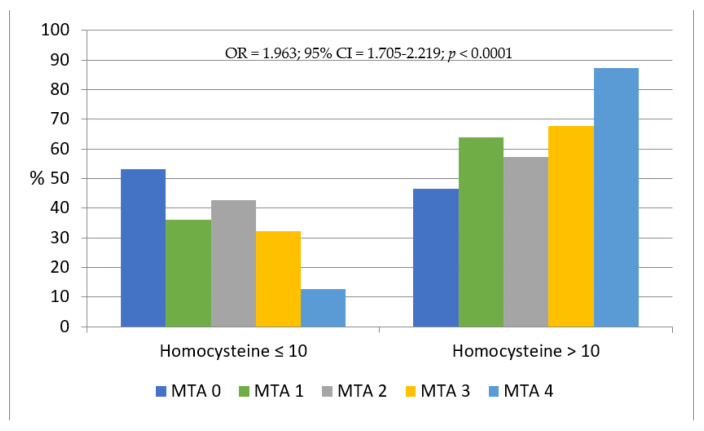
Distribution of MTA score according to homocysteine levels. Legend: **MTA**, Medial Temporal Lobe Atrophy; **%**, percentage of homocysteine level distributions.

**Figure 2 biomedicines-10-01381-f002:**
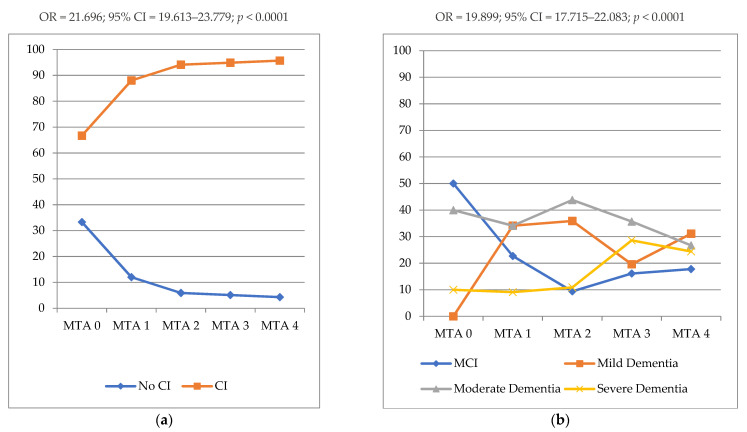
Distribution of MTA score according to the cognitive impairment presence/absence (**a**) and severity (**b**). Legend: **MTA**, Medial Temporal Lobe Atrophy; **NoCI**, No Cognitive Impairment; **CI**, Cognitive Impairment; **MCI**, Mild Cognitive Impairment; **%**, percentage of cognitive impairment presence/absence (**a**) and severity (**b**) distributions.

**Figure 3 biomedicines-10-01381-f003:**
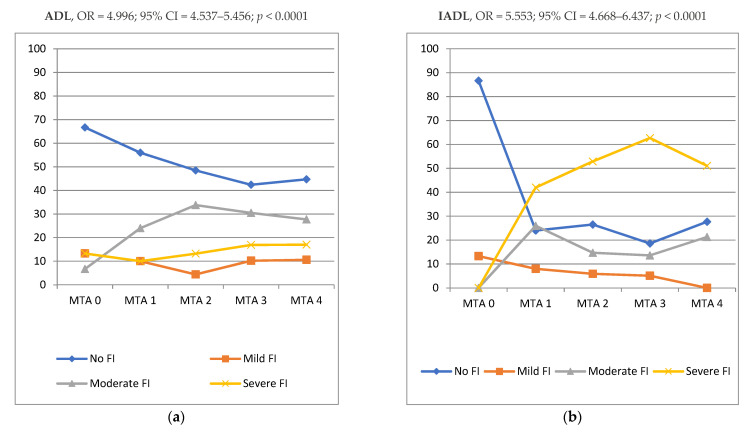
Distribution of MTA score according to the functional impairment (FI) severity evaluated by Activity of Daily Living (**a**) and Instrumental Activity of Daily Living (**b**). Legend: **MTA**, Medial Temporal Lobe Atrophy; **ADL**, Activities of Daily Living; **IADL**, Instrumental Activities of Daily Living.

**Table 1 biomedicines-10-01381-t001:** Demographic, cognitive, functional, clinical, and biochemical characteristics of older patients according to MTA score.

	MTA 0 n = 14	MTA 1 n = 44	MTA 2 n = 63	MTA 3 n = 79	MTA 4 n = 39	*p*-Value
**Sex**						**0.025**
Males/Females	7/7	22/22	22/41	25/54	16/23	
Males (%)	50.0	50.00	34.90	31.6	41.0	
**Age** (years)						**<0.0001**
Mean ± SD	72.60 ± 9.30	78.10 ± 6.57	79.12 ± 6.21	81.08 ± 6.19	80.70 ± 6.55	
Range	55.00–87.00	64.00–91.00	53.00–90.00	69.00–97.00	62.00–95.00	
**MMSE**						**<0.0001**
Mean ± SD	22.80 ± 7.39	19.91 ± 6.23	17.52 ± 6.48	14.94 ± 8.34	14.90 ± 8.80	
Range	9.00–30.00	5.00–30.00	0–30.00	0–28.00	0–27.00	
**FAB**						**0.017**
Mean ± SD	11.80 ± 6.03	10.68 ± 5.69	8.71 ± 5.74	7.38 ± 6.01	9.00 ± 5.76	
Range	0 – 18.00	0 – 18.00	0 – 18.00	0 – 18.00	0–18.00	
**ADL**						**<0.0001**
Mean ± SD	5.13 ± 1.51	4.88 ± 1.49	4.52 ± 1.65	4.39 ± 1.67	4.38 ± 1.84	
Range	2.00–6.00	1.00–6.00	1.00–6.00	1.00–6.00	0–6.00	
**IADL**						**<0.0001**
Mean ± SD	5.60 ± 3.27	4.14 ± 3.04	3.68 ± 3.25	2.92 ± 3.10	2.87 ± 3.43	
Range	0–8.00	0–8.00	0–8.00	0–8.00	0–8.00	
**Diagnosis**						**<0.0001**
No Cognitive Impairment	4 (28.6)	7 (15.9)	0	0	0	
Mild Cognitive Impairment	2 (14.3)	9 (20.5)	5 (7.9)	0	0	
Alzheimer’s disease	0	2 (4.5)	22 (34.9)	36 (45.6)	26 (66.7)	
Vascular Dementia	1 (7.1)	10 (22.7)	20 (31.7)	23 (29.1)	8 (20.5)	
Psycho-behavioral symptoms	7 (50.0)	16 (36.4)	16 (25.4)	20 (25.3)	5 (12.8)	
**Homocysteine, μmol/L**						**0.032**
Mean ± SD	9.61 ± 2.95	11.95 ± 3.65	13.05 ± 7.39	14.19 ± 7.37	14.44 ± 5.07	
Range	5.90–15.00	64.20–23.00	6.00–47.00	6.30–49.88	8.10–31.00	
**Hypertension**						
Yes–n (%)	9 (64.3)	28 (63.6)	38 (60.3)	48 (60.8)	24 (61.5)	0.996
No–n (%)	5 (35.7)	16 (36.4)	25 (39.7)	31 (39.2)	15 (38.5)	
**Diabetes**						
Yes–n (%)	2 (14.3)	9 (20.5)	17 (27.0)	26 (32.9)	12 (30.8)	0.450
No–n (%)	12 (85.7)	35 (79.5)	46 (73.0)	53 (67.1)	27 (69.2)	
**Atrial fibrillation**						
Yes–n (%)	2 (14.3)	3 (6.8)	8 (12.7)	6 (7.6)	3 (7.7)	0.735
No–n (%)	12 (85.7)	41 (93.2)	55 (87.3)	73 (92.4)	36 (92.3)	
**Dyslipidemia**						
Yes–n (%)	7 (50.0)	19 (43.2)	19 (30.2)	25 (31.6)	12 (30.8)	0.404
No–n (%)	7 (50.0)	25 (56.8)	44 (69.8)	54 (68.4)	27 (69.2)	

Legend: **MTA**, Medial Temporal Lobe Atrophy; **MMSE**, Mini-Mental State Examination; **FAB**, Frontal Assessment Battery; **ADL**, Activities of Daily Living; **IADL**, Instrumental Activities of Daily Living. The *p*-values in bold represent the significant values (<0.05).

**Table 2 biomedicines-10-01381-t002:** Association of MTA score and NPI domains, adjusted by cognitive impairment presence/absence and severity.

	MTA 0 n = 15	MTA 1 n = 50	MTA 2 n = 68	MTA 3 n = 59	MTA 4 n = 47	*p*-Value	OR	95% CI
**NPI Total score**								
Mean ± SD	19.33 ± 19.52	18.44 ± 15.45	19.30 ± 18.32	21.83 ± 17.17	22.44 ± 16.56	**0.002**	11.564	4.452–18.676
Range	0–58.00	0–54.00	0–61.00	0–54.00	0–58.00			
**Delusion**								
Mean ± SD	0.87 ± 2.47	0.44 ± 1.85	0.31 ± 1.38	0.53 ± 2.05	0.60 ± 2.07	0.879	0.068	−0.808–0.944
Range	0–9.00	0–9.00	0–9.00	0–9.00	0–9.00			
**Hallucination**								
Mean ± SD	0	0.76 ± 2.20	0.87 ± 2.46	0.90 ± 2.56	0.53 ± 2.01	0.839	0.107	−0.931–1.145
Range	0	0–9.00	0–9.00	0–9.00	0–9.00			
**Agitation/Aggression**								
Mean ± SD	1.93 ± 3.17	1.72 ± 2.70	2.50 ± 3.33	3.80 ± 4.11	3.21 ± 4.06	0.296	0.836	−0.735–2.407
Range	0–9.00	0–9.00	0–12.00	0–12.00	0–12.00			
**Depression**								
Mean ± SD	2.51 ± 3.39	3.25 ± 3.49	4.20 ± 4.65	4.24 ± 3.93	5.03 ± 4.12	**<0.0001**	4.869	3.053–6.686
Range	0–12.00	0–12.00	0–12.00	0–12.00	0–12.00			
**Anxiety**								
Mean ± SD	1.62 ± 3.11	1.75 ± 3.24	2.08 ± 3.37	2.93 ± 3.58	3.19 ± 4.02	**0.001**	2.792	1.120–4.464
Range	0–9.00	0–12.00	0–9.00	0–9.00	0–12.00			
**Euphoria**								
Mean ± SD	0	0	0.07 ± 0.50	0.17 ± 1.18	0.09 ± 0.58	0.824	−0.037	−0.363–0.289
Range	0	0	0–4.00	0–9.00	0–4.00			
**Apathy/Indifference**								
Mean ± SD	3.47 ± 4.66	2.88 ± 4.22	3.01 ± 3.88	4.12 ± 4.37	3.45 ± 4.00	0.366	0.815	−0.955–2.585
Range	0–12.00	0–12.00	0–12.00	0–12.00	0–12.00			
**Disinhibition**								
Mean ± SD	0	0.16 ± 0.79	0.65 ± 2.04	0.32 ± 1.65	0.19 ± 1.31	0.932	0.031	−0.692–0.754
Range	0	0–4.00	0–9.00	0–9.00	0–9.00			
**Irritability/Lability**								
Mean ± SD	2.13 ± 3.70	1.22 ± 2.79	2.47 ± 3.54	3.14 ± 3.78	2.85 ± 3.95	0.898	0.103	−1.470–1.675
Range	0–9.00	0–9.00	0–9.00	0–9.00	0–12.00			
**Aberrant Motor Behavior**								
Mean ± SD	0.27 ± 1.03	0.56 ± 2.01	0.50 ± 1.76	0.53 ± 2.05	0.09 ± 0.58	0.305	0.413	−0.378–1.204
Range	0–4.00	0–9.00	0–9.00	0–9.00	0–4.00			
**Sleep/night-time behavior**								
Mean ± SD	2.67 ± 4.08	3.10 ± 3.92	2.94 ± 3.94	2.59 ± 3.87	3.23 ± 3.93	0.333	0.858	−0.884–2.600
Range	0–9.00	0–12.00	0–12.00	0–12.00	0–12.00			
**Appetite/eating change**								
Mean ± SD	0.87 ± 2.48	1.20 ± 2.63	0.68 ± 1.89	0.75 ± 2.53	0.85 ± 2.39	0.209	0.704	−0.398–1.805
Range	0–9.00	0–9.00	0–9.00	0–12.00	0–12.00			

Legend: **MTA**, Medial Temporal Lobe Atrophy; **NPI**, Neuropsychiatric Inventory. The *p*-values in bold represent the significant values (<0.05).

## Data Availability

Not applicable.

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
