# Peer review of "CT-Detected MTA Score Related to Disability and Behavior in Older People with Cognitive Impairment"

_biomedicines, 2022, doi:10.3390/biomedicines10061381_

Round 1

Reviewer 1 Report

The authors of the work “MTA score related to disability and behavioural in elderly people with cognitive impairment” investigated the relationship between MTA score and functional impairment, cognitive deficit, and psycho-behavioral disorders. This study is interesting and encourages to use MTA in neurodegenerative diagnosis.

However, some minor points require a review:

  1. In the Introduction section, more recent references should be added.
  2. In the Discussion section, more explanation is required about the specificity of the results (MTA) in AD cases, in comparison with other neurodegenerative diseases, such as vascular disease.

Also, more discussion regarding psycho-behavioral results with recent references is required.

  1. Some limitations should be added in the last paragraph, such as, the lack of CSF biomarkers to classify biologically the AD and non-AD participants;

Author Response

  1. The authors of the work “MTA score related to disability and behavioural in elderly people with cognitive impairment” investigated the relationship between MTA score and functional impairment, cognitive deficit, and psycho-behavioral disorders. This study is interesting and encourages to use MTA in neurodegenerative diagnosis. However, some minor points require a review.                                                                                            1. We thank the reviewer for his/her positive and constructive observations. Below, please find item-by-item responses to your comments, which are included verbatim.

  1. In the Introduction section, more recent references should be added.        2. According to reviewer suggestion, we added a recent reference [1] in the text. As regards the other references, these are essential to clarify the background from which our study started.

  1. In the Discussion section, more explanation is required about the specificity of the results (MTA) in AD cases, in comparison with other neurodegenerative diseases, such as vascular disease. Also, more discussion regarding psycho-behavioral results with recent references is required.
  1. In Discussion section, we added the comparisons with other neurodegenerative diseases, and a more detailed discussion about psycho-behavioral results.

  1. Some limitations should be added in the last paragraph, such as, the lack of CSF biomarkers to classify biologically the AD and non-AD participants.  4. The aforesaid limitation has been added, according to reviewer observation.

We thank the reviewer again for his/her suggestions that have certainly improved and enriched our work.

Reviewer 2 Report

In this article, the authors, provide quite good evidence that brain CT can be used in MTA scoring. The scoring also is positively related to cognitive and behavioural impairment.

Some recommendations to authors:

1) In title you should mention the use of brain CT, as the main tool for the MTA scoring system

2) There are not anywhere the parameters of the brain CT. Any brain CT may not be suitable for scoring. Anyhow in the original Schelten’s article, coronal slices were used for scoring. I think you need to provide some more details for the brain CT protocol.

3) There are some patients with ‘presence of neuropsychiatric symptoms without dementia’. This is not a well-defined group. Have the authors excluded patients with prior mental illnesses (depression, bipolar disorder or schizophrenia)? Are these behavioral symptoms emerged after the age of 55 years old (the term of Mild Behavioral Impairment has emerged last decade)? Do they have any minor cognitive issues (eg executive disorder), and they can grouped under an MCI non_AD group? Could these symptoms have caused by vascular changes (eg Vascular Impairment- non reaching the status of VAD)? Could these patients experience the precipitating behavioral symptoms of Alzheimer’s or Parkinson’s disease? Please define this group better and explain why you have included these patients.

3) I encourage authors to define in more details the cause they have chosen homocysteine levels as vascular marker. Do the authors speculate why there is a relation between homocysteine levels and the MTA scoring severity?

4) Please provide some more bibliography implicating MTA to depression and anxiety in degenerative diseases

Author Response

  1. In this article, the authors provide quite good evidence that brain CT can be used in MTA scoring. The scoring also is positively related to cognitive and behavioural impairment. Some recommendations to authors:
  1. We thank the reviewer for his/her positive and constructive observations. Below, please find item-by-item responses to your comments, which are included verbatim.

  1. In title you should mention the use of brain CT, as the main tool for the MTA scoring system. There are not anywhere the parameters of the brain CT. Any brain CT may not be suitable for scoring. Anyhow in the original Schelten’s article, coronal slices were used for scoring. I think you need to provide some more details for the brain CT protocol.
  1. According to the reviewer observations, we added the follow title “CT-detected MTA score related to disability and behaviour in elderly people with cognitive impairment” and more details about CT in section 2.3.

  1. There are some patients with ‘presence of neuropsychiatric symptoms without dementia’. This is not a well-defined group. Have the authors excluded patients with prior mental illnesses (depression, bipolar disorder or schizophrenia)? Are these behavioral symptoms emerged after the age of 55 years old (the term of Mild Behavioral Impairment has emerged last decade)? Do they have any minor cognitive issues (eg executive disorder), and they can grouped under an MCI non_AD group? Could these symptoms have caused by vascular changes (eg Vascular Impairment- non reaching the status of VAD)? Could these patients experience the precipitating behavioral symptoms of Alzheimer’s or Parkinson’s disease? Please define this group better and explain why you have included these patients.                                                                                                         3. The finding of a reduction in hippocampal volume is also associated with psycho-behavioral disorders regardless of the cognitive deficit, as already described in the text.

  1. I encourage authors to define in more details the cause they have chosen homocysteine levels as vascular marker. Do the authors speculate why there is a relation between homocysteine levels and the MTA scoring severity?                                                                                                         4. Homocysteine was recently studied by us (you can see the reference n. 66). It is a sensitive, specific and easily repeatable bio-humoral marker. MTA is an index of atrophy which we link with cognitive impairment. And homocysteine showed a strong relationship with the severity of cognitive and functional deficits. These information were already added in the text.

  1. Please provide some more bibliography implicating MTA to depression and anxiety in degenerative diseases.                                                           5. We added more detailed discussion about psycho-behavioral results, according to reviewer suggestion.

We thank the reviewer again for his/her suggestions that have certainly improved and enriched our work.

Reviewer 3 Report

Lauriola et al are presenting a manuscript investigating the use of MTA scoring from CT scans in investing patient characteristics related to cognition, disability, biochemical measures. The paper adds to the evaluation of MTA using CT scans. 

  1. Revise the instructions for authors again and reformat the manuscript according to these instructions. 
  2. Please carefully revise the text in detail and perform a thorough proofreading. Moreover, there are many typos and words out of place that make it harder for the reader to follow the text. Here are a few:

Throughout the text, abbreviations are explained differently, ie the first letter in each abbreviated word is written in capital letters, or lower case letters. An example: ADL: "Activities of Daily Living" or "activities or daily living". Please use ONE method in the text and the tables and figure legends.  

Line 1: Title is incomplete, "behavioural" is missing a noun or can be made into a noun

Line 14: Remove all headings from the abstracts 

Line 15: Add that CT scans were used

Line 16-17: The number of significant digits can be reduced when describing the ages here and in the rest of the text

Line 20: Capital H on hypertension

Line 38: Add space before In recent years

Line 40: Restructure the sentence staring with Performed the...

Line 48: Add space before AD

Line 51: Add space after (CT)

Line 78: Add "from CT scans" after "MTA score"

Line 127: Remove "From"

Section 2.2.: Add a description on how the CT scan was performed

Section 2.3.: Expand the description on how the MTA was performed (number of persons analyzing, was the analysis done blindly etc)

Line 143: Please write the M&M in past tense, ie use "atrophy was evaluated" and in the rest of this paragraph

Line 147: "Value" can be removed

Line 148: Remove "Diagnosis"

Line 151: Better describe "in full auto analyzer". Model, process?

Line 152 and in other parts of the text: Homocysteine is randomly spelled with capital H in some locations. Change this throughout the text to lower case h

Line 154: Remove space before "cholesterol"

Line 173. Remove "This" and "the" before "inclusion"

Line 174: 60 years should probably read 50 years

Line 175: Remove space before "Thus"

Line 175: Different value digits are used for percentage of men and women. Please adjust

Lines 177-182: This information is already found in Table 1. Can be removed here in the text

Line 192: Add lower case h at homocysteine

Lines 197 and 198: Explain the abbreviations CI and NoCI. Also add space before CI

Pleases insert tables and figures in the order they appear

Table 1: Please format the table/move on the page so that the table fits on one full page. Also use either capital or lower case letters for in a consistent matter (now it is varying, for example among the diagnoses and in the legend. In addition, n= is written with both capital N and lower case n)

Table 2: Please see comments for Table 1. NPI can be removed from the domain names. In title, remove space before MTA

Figure 1: Revise this figure: misspelled homocysteine, give a unit to 10 (the levels), explain what % refers to. In legend: ad lower case h on homocysteine. 

Figure 2: Add A and B to the two graphs. Explain what % corresponds to (% of ...). Legend: add space in No CI

Line 221: Can this text be written in larger font or as part of the images to stand out better?

Figure 3: Please see comments for Figure 2

Discussion:

In its current state, the Discussion is degraded into several paragraphs only consisting of one sentence each. Please revise the text to help the text flow better. 

Please expand the discussion to comparisons between outcomes from MTA scoring from CT and MRI scans. 

Please include a discussion on how MTA scoring from CT scans can be used clinically in the diagnostic and care of patients seeking care at primary care units or specialist units. 

Please include a section on Future research

Author Response

  1. Lauriola et al are presenting a manuscript investigating the use of MTA scoring from CT scans in investing patient characteristics related to cognition, disability, biochemical measures. The paper adds to the evaluation of MTA using CT scans. Revise the instructions for authors again and reformat the manuscript according to these instructions.
  1. We thank the reviewer for his/her positive and constructive observations. Below, please find item-by-item responses to your comments, which are included verbatim.

  1. Please carefully revise the text in detail and perform a thorough proofreading. Moreover, there are many typos and words out of place that make it harder for the reader to follow the text. Here are a few: Throughout the text, abbreviations are explained differently, ie the first letter in each abbreviated word is written in capital letters, or lower case letters. An example: ADL: "Activities of Daily Living" or "activities or daily living". Please use ONE method in the text and the tables and figure legends. 
  1. We correct in Activities of Daily Living.

  1. Line 1: Title is incomplete, "behavioural" is missing a noun or can be made into a noun.                                                                                                   3. According to reviewer observation, we changed “behavioural” in “behavior”.

  1. Line 14: Remove all headings from the abstracts. Line 15: Add that CT scans were used. Line 16-17: The number of significant digits can be reduced when describing the ages here and in the rest of the text. Line 20: Capital H on hypertension. Line 38: Add space before In recent years. Line 40: Restructure the sentence staring with Performed the... Line 48: Add space before AD. Line 51: Add space after (CT). Line 78: Add "from CT scans" after "MTA score". Line 127: Remove "From"
  1. According to reviewer suggestions, all aforesaid correction have been made.

  1. Section 2.2.: Add a description on how the CT scan was performed.           5. We added the CT scan description in Section 2.3., as suggested by the reviewer.

  1. Section 2.3.: Expand the description on how the MTA was performed (number of persons analyzing, was the analysis done blindly etc).               6. We expanded the description about MTA, as suggested by the reviewer.

  1. Line 143: Please write the M&M in past tense, ie use "atrophy was evaluated" and in the rest of this paragraph. Line 147: "Value" can be removed. Line 148: Remove "Diagnosis".
  1. According to reviewer suggestions, all aforesaid correction have been made.

  1. Line 151: Better describe "in full auto analyzer". Model, process?                8. It is a model of machine. It consists of a tray where the samples are loaded to be tested. In the text, we added “in full auto analyzer machine” in order to clarify.

  1. Line 152 and in other parts of the text: Homocysteine is randomly spelled with capital H in some locations. Change this throughout the text to lower case h. Line 154: Remove space before "cholesterol". Line 173. Remove "This" and "the" before "inclusion". Line 174: 60 years should probably read 50 years. Line 175: Remove space before "Thus". Line 175: Different value digits are used for percentage of men and women. Please adjust Lines 177-182: This information is already found in Table 1. Can be removed here in the text. Line 192: Add lower case h at homocysteine.
  1. According to reviewer suggestions, all aforesaid correction have been made.

  1. Lines 197 and 198: Explain the abbreviations CI and NoCI. Also add space before CI.                                                                                                     10. About CI, the explanation has been added at line 93. About NoCI, the explanation was just at Line 97.

  1. Pleases insert tables and figures in the order they appear.                         11. We ordered the tables and figures, as reviewer suggested.
  2. Table 1: Please format the table/move on the page so that the table fits on one full page. Also use either capital or lower case letters for in a consistent matter (now it is varying, for example among the diagnoses and in the legend. In addition, n= is written with both capital N and lower case n).                                                                                                                 12. According to reviewer suggestions, the correction have been made.

  1. Table 2: Please see comments for Table 1. NPI can be removed from the domain names. In title, remove space before MTA.                                    13. According to reviewer suggestions, the correction have been made except for fitting in a full page.

  1. Figure 1: Revise this figure: misspelled homocysteine, give a unit to 10 (the levels), explain what % refers to. In legend: ad lower case h on homocysteine.                                                                                             14. The Figure 1 has been revised, according to reviewer observations.

  1. Figure 2: Add A and B to the two graphs. Explain what % corresponds to (% of ...). Legend: add space in No CI.                                                         15. The Figure 2 has been revised, according to reviewer observations.

  1. Line 221: Can this text be written in larger font or as part of the images to stand out better?                                                                                         16. Yes, it was been made.

  1. Figure 3: Please see comments for Figure.                                                  17. The Figure 3 has been revised, according to reviewer observations.

  1. Discussion: In its current state, the Discussion is degraded into several paragraphs only consisting of one sentence each. Please revise the text to help the text flow better. Please expand the discussion to comparisons between outcomes from MTA scoring from CT and MRI scans. Please include a discussion on how MTA scoring from CT scans can be used clinically in the diagnostic and care of patients seeking care at primary care units or specialist units. Please include a section on Future research
  1. We revised the Discussion section, expanding the text about comparison MTA score from CT and MRI scan, and about the use of CT-detected MTA score in clinical practice.

We thank the reviewer again for his/her suggestions that have certainly improved and enriched our work.